# Stochastic Policy Optimization with Heuristic Information for Robot Learning

**Seonghyun Kim,  Ingook Jang,  Samyeul Noh,  Hyunseok Kim**[*]
Electronics and Telecommunications Research Institute
218 Gajeong-ro, Daejeon, Korea
{kim-sh, ingook, samuel, hertzkim}@etri.re.kr

**Abstract:** Stochastic policy-based deep reinforcement learning (RL) approaches have remarkably succeeded to deal with continuous control tasks. However, applying these methods to manipulation tasks remains a challenge since actuators of a robot manipulator require high dimensional continuous action spaces. In this paper, we propose exploration-bounded exploration actor-critic (EBE-AC), a novel deep RL approach to combine stochastic policy optimization with interpretable human knowledge. The human knowledge is defined as heuristic information based on both physical relationships between a robot and objects and binary signals of whether the robot has achieved certain states. The proposed approach, EBE-AC, combines an off-policy actor-critic algorithm with an entropy maximization based on the heuristic information. On a robotic manipulation task, we demonstrate that EBE-AC outperforms prior state-of-the-art off-policy actor-critic deep RL algorithms in terms of sample efficiency. In addition, we found that EBE-AC can be easily combined with latent information, where EBE-AC with latent information further improved sample efficiency and robustness.

**Keywords:** Robot manipulation, Reinforcement learning, Stochastic policy

## 1   Introduction

A challenge to human-level AI [1] is still ongoing research area and recently deep reinforcement learning (RL) approaches have remarkably shown outstanding results beyond human in the fields of board and video games [2, 3]. As a greatly unexplored area which adapts deep RL, robot manipulation is one of the challenging applications due to complex nonlinear dynamics and huge continuous action spaces [4]-[7]. Various and stable robot manipulation skills in real-world environments [8], where the laws of physics are strictly applied, are required to solve complex tasks, such as moving, grasping, pushing, and placing. It is, therefore, still a challenge to solve complex tasks through deep RL approaches.

In front of the advanced dexterity capabilities of human hands, learning the dexterity with functionality to autonomously pick and place objects has always been a long-standing challenge for robotics [9, 10]. To solve complex manipulation tasks, various approaches have been presented to utilize human knowledge, which includes learning from human demonstrations and optimizing hyper-parameters empirically [11]-[18]. The first improves sample efficiency of learning by providing good expert demonstration data to solve a given task [13]-[16]. In general, the human demonstrations can be used not only as a loss function of inverse RL but also as reusable experience data of off-policy RL. However, it is expensive to obtain real-world demonstrations for every single task.

As the second approach utilizing human knowledge, empirical hyper-parameter tuning [19], has helped optimize learning progress since hyper-parameters are highly related to not only neural network architectures but also reward designs in the deep RL. On the whole, hyper-parameter tuning is widely used to ensure that learning algorithms gain good performance, however, it is vulnerable to initial hyper-parameter settings for each task [18]. This results in the inefficiency of the learning process because it is necessary to perform hyper-parameter tuning every task.

---

[*]corresponding author

5th Conference on Robot Learning (CoRL 2021), London, UK.

A heuristic (informed) strategy using problem-specific knowledge has been generally known as a more efficient solution than an uninformed strategy using only the problem itself [20]. Since manipulation skills are under the laws of physics, informed human knowledge, such as moving in the direction of narrowing the distance, being in alignment with the target object, and grasping the center of gravity of the object, can be helpful to find solutions efficiently in robot manipulation.

In this paper, we propose a novel deep RL approach to combine stochastic policy optimization with human knowledge, which is exploration-bounded exploration actor-critic (EBE-AC). EBE-AC applies interpretable and reusable human knowledge, heuristic information based on both physical relationships between a robot and objects and binary signals of whether the robot has achieved certain states, to an off-policy actor-critic algorithm. The main contributions of this paper are summarized as follows:

- We design a novel information theoretic metric in order for EBE-AC to have dependencies between actions and heuristic information.

- We propose an $\epsilon$-exploration method using two types of stochastic policies, exploration and bounded exploration, in order to store diverse experiences in a shared replay buffer.

- We propose a temperature optimization for given entropies corresponding to the two policies, in order to balance between the two.

To verify the performance of EBE-AC, we conducted experiments on a robotic manipulation task, *Pick-and-Lift*. The experimental results show that EBE-AC outperforms prior state-of-the-art off-policy actor-critic deep RL algorithms in terms of sample efficiency. In addition, we found that EBE-AC can be easily combined with latent information as DIAYN [21] does, where EBE-AC with latent information further improved sample efficiency and robustness.

## 2  Related Work

Due to the difficulty of obtaining stable and steady improvement of deterministic policy in applying complex and high dimensional tasks, stochastic policy optimization based on information theory has been examined in problems with continuous state-action spaces [22]-[24]. Many works show that applying a stochastic policy's entropy to an objective function results in good performances. In particular, regularizing both policy and value functions with entropy shows the convergence and stability improvements during training processes.

Through maximum entropy RL [23], a policy is encouraged to perform diverse actions to widely explore a state space while maximizing expected returns. Owing to the diverse actions being able to maximize expected returns, the learned policy can have multiple optimal solutions and obtain stable performance regardless of random initial states. In addition, the maximum entropy approaches are also robust to environmental disturbances such as the presence of adversaries and dynamics change over time because these works focus on the entropy of the action distribution for a given state.

In the unsupervised emergence of diverse skills dictating the states [21], the mutual information between states and latent variables is leveraged to autonomously discover diverse skills. To ensure the different skills being distinguishable, the mutual information between actions and skills should be minimized for a given state. By utilizing the mutual information as a metric of the objective function, a stochastic policy can be also stably and steadily improved under the consideration of additional latent variables.

These prior works utilize rich exploration to obtain various experiences at the beginning of learning, and tend to reduce the exploration as the learning progresses in order to acquire good expected returns stably. However, when a policy is stuck in inevitable local optimums which must be passed to complete a given complex task, the policy needs to utilize rich exploration in order to escape the local optimums even in the middle of learning. In other words, it is necessary that a policy simultaneously pursues rich exploration and stable expected returns throughout the learning period. This paper proposes a novel way to combine human knowledge with a stochastic policy by leveraging the information theoretic metrics. Unlike reducing the mutual information between actions and latent variables in [21], the proposed method increases the mutual information between actions and heuristic information based on human knowledge. As a result, the dependency between actions and heuristic information will be guaranteed and experiences obtained by actions are related to heuristic

information. Based on the metric, $\epsilon$-exploration method using two types of stochastic policies is proposed in order to maintain rich exploration throughout the learning period. Based on the proposed method, the sample efficiency is also improved and the stochastic policy achieves the robustness to complex tasks.

## 3   Preliminaries

### 3.1   Notation

A Markov decision process is defined by the tuple $(\mathcal{S}, \mathcal{A}, p, r)$, where $\mathcal{S}$ is a state space, $\mathcal{A}$ is a continuous action space, $p : \mathcal{S} \times \mathcal{A} \times \mathcal{S} \rightarrow \mathbb{R}_+$ is a transition probability distribution, and $r : \mathcal{S} \times \mathcal{A} \rightarrow \mathbb{R}$ is a reward function. In episodic tasks, a discounted return at time $t$ is given as $G_t = \sum_{k=t+1}^{T} \gamma^{k-t-1} r(s_k, a_k)$, where $T$ is an episode horizon, $\gamma$ is a discount factor, and $r(s_k, a_k)$ is a reward for the state-action pair $(s_k, a_k)$. A value function for a policy $\pi(a_t|s_t)$ is represented as $v_\pi(s_t) = \mathbb{E}_\pi[G_t]$.

### 3.2   Objective function

We consider combining RL with the information theoretic metrics, which can encourage policy optimization in both ways to diverse actions and specific actions in a given state. Therefore, we propose an objective function to learn a policy conditioned on heuristic information according to human knowledge, where the policy can select actions considering not only the states but also the heuristic information.

First, we define an information theoretic reward as

$$\widetilde{r}(s_t, a_t) = r(s_t, a_t) + w_s I(a_t; z_t|s_t) + w_z h_{\rho_z}(a_t|s_t, z_t) \tag{1}$$
$$= r(s_t, a_t) + w_s\{h_{\rho_s}(a_t|s_t) - h_{\rho_z}(a_t|s_t, z_t)\} + w_z h_{\rho_z}(a_t|s_t, z_t), \tag{2}$$

where $z_t$ is heuristic information, $I(a_t; z_t|s_t)$ is a mutual information between $a_t$ and $z_t$ for the given $s_t$, $h_\rho(\cdot)$ is an entropy for the distribution $\rho$, $w_s$ and $w_z$ are positive temperatures, and $\rho_s$ and $\rho_z$ are distributions of action $a_t$ conditioned on $s_t$ and $(s_t, z_t)$, respectively. The distributions $\rho_s$ and $\rho_z$ are induced by a policy $\pi$. The heuristic information $z_t$ is defined by external function $z_t = f(s_t)$ based on $s_t$, where $f(\cdot)$ can be physical information between a robot and objects in addition to binary signals representing specific states.[2]

Maximizing the reward $\widetilde{r}(s_t, a_t)$ in Equation (1) means that a policy has to find an optimal solution to maximize the reward from the environment while guaranteeing high relation between $a_t$ and $z_t$, and diverse actions for given information $s_t$ and $z_t$. In episodic tasks, a discounted return for $\widetilde{r}(s_t, a_t)$ is derived as

$$\widetilde{G}_t = \sum_{k=t+1}^{T} \gamma^{k-t-1} \widetilde{r}(s_k, a_k) = G_t + w_s H_{\rho_s}(a_t|s_t) + (w_z - w_s) H_{\rho_z}(a_t|s_t, z_t), \tag{3}$$

where $H_\rho(\cdot) = \sum_{k=t+1}^{T} \gamma^{k-t-1} h_\rho(\cdot)$ is a discounted sum of entropy.

By using the discounted return in Equation (3), the objective function is defined as follows:

$$J_\pi(s_t) = \mathbb{E}_\pi[\widetilde{G}_t] = v_\pi(s_t) + w_s \overline{H}_{\rho_s}(a_t|s_t) + (w_z - w_s) \overline{H}_{\rho_z}(a_t|s_t, z_t), \tag{4}$$

where $\overline{H}_\rho(\cdot) = \mathbb{E}_\pi[H_\rho(\cdot)]$.

To maximize the objective function in Equation (4) with a simple and fast way to implement, this paper designs $\epsilon$-exploration based policy using two types of stochastic policies as follows:[3]

$$\pi = \pi_m = \begin{cases} \pi_s(a_t|s_t), & \text{for } p_m(s) = 1 - \epsilon \\ \pi_z(a_t|s_t, z_t), & \text{for } p_m(z) = \epsilon \end{cases}, \tag{5}$$

---

[2]Detailed examples of heuristic information are described in Appendix D.3.

[3]In comparison with an $\epsilon$-greedy method which uses policy-based actions and random actions with $\epsilon$ probability for exploration effects, the $\epsilon$-exploration method uses two types of policies having different exploration characteristics measured in entropies where one learns to maximize a return with rich explorations and the other learns to maximize a return with bounded explorations.

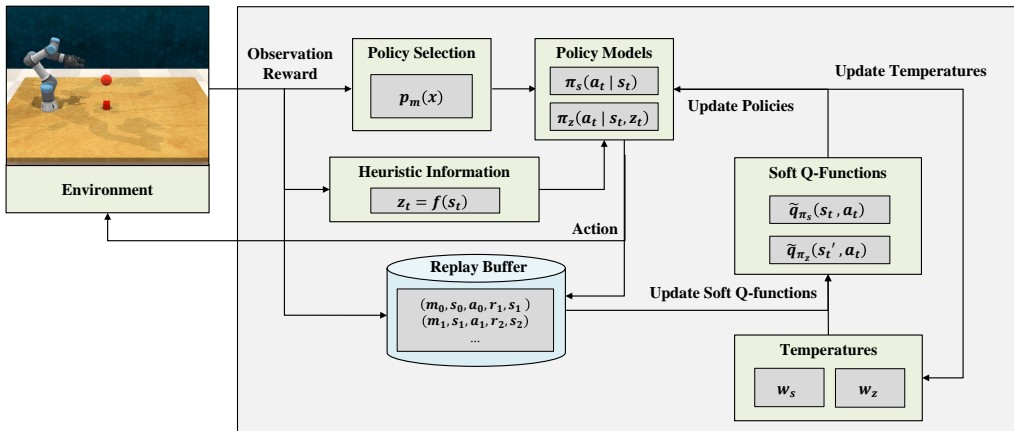

Figure 1: **EBE-AC Algorithm:** EBE-AC algorithm consists of two types of policies. Each policy shares the replay buffer and updates its own soft Q-function in an off-policy soft actor-critic(SAC) manner. After updating the soft Q-function and policies, the temperatures are updated.

where $p_m(x) = P(m = x)$ is a probability mass function for selecting policy $\pi_x$, and $\epsilon$ is a positive constant. By Equation (5), the objective function in Equation (4) is reformulated as follows:

$$
\begin{aligned}
J_{\pi_m}(s_t) &= \mathbb{E}_{m \in \{s,z\}}[v_{\pi_m}(s_t) + \overline{H}_{\pi_m}] \\
&= (1 - \epsilon)\{v_{\pi_s}(s_t) + w_s \overline{H}_{\pi_s}(a_t|s_t)\} \\
&\quad + \epsilon\{v_{\pi_z}(s_t) + (w_z - w_s)\overline{H}_{\pi_z}(a_t|s_t, z_t)\} = J_{\pi_s, \pi_z}(s_t),
\end{aligned}
\tag{6}
$$

where

$$
\overline{H}_{\pi_m} = \begin{cases} w_s \overline{H}_{\pi_s}(a_t|s_t), & \text{for } m = s \\ (w_z - w_s)\overline{H}_{\pi_z}(a_t|s_t, z_t), & \text{for } m = z \end{cases}.
\tag{7}
$$

Thus, the maximization problem for Equation (6) is represented as

$$
\max_{\pi_s, \pi_z} \; J_{\pi_s, \pi_z}(s_t), \quad \forall s_t \in \mathcal{S}.
\tag{8}
$$

## 4 Exploration-Bounded Exploration Actor-Critic

### 4.1 Problem decomposition

The maximization problem in Equation (8) is decomposed into two sub-problems: a maximization problem with exploration entropy and a maximization problem with bounded exploration entropy for the policies $\pi_s$ and $\pi_z$, respectively.

#### 4.1.1 Learning with exploration entropy

For the policy $\pi_s$, the problem in Equation (8) is reduced as

$$
\max_{\pi_s} \; J_{\pi_s}(s_t), \quad \forall s_t \in \mathcal{S},
\tag{9}
$$

where $J_{\pi_s}(s_t) = v_{\pi_s}(s_t) + w_s \overline{H}_{\pi_s}(a_t|s_t)$. By representing $J_{\pi_s}(s_t) = \widetilde{v}_{\pi_s}(s_t)$, Equation (9) can be derived as follows:

$$
J_{\pi_s}(s_t) = \widetilde{v}_{\pi_s}(s_t) = \mathbb{E}_{\pi_s}[\widetilde{q}_{\pi_s}(s_t, a_t) + w_s h_{\pi_s}(a_t|s_t)]
\tag{10}
$$

where $\widetilde{q}_{\pi_s}(s_t, a_t) = r(s_t, a_t) + \gamma \mathbb{E}_{s_{t+1} \sim p}[\widetilde{v}_{\pi_s}(s_{t+1})]$ is the soft Q-function for $\pi_s$ [23].[4] Because the entropy term $w_s h_{\pi_s}(a_t|s_t)$ incentivizes the policy to choose actions with a high variance, the policy learns to be exploratory for given states.

---

[4]Detailed derivations of Equation (10) are described in Appendix A.

### 4.1.2 Learning with bounded exploration entropy

For the policy $\pi_z$, the problem in Equation (8) is reduced as

$$\max_{\pi_z} \; J_{\pi_z}(s_t), \quad \forall s_t \in \mathcal{S}, \tag{11}$$

where $J_{\pi_z}(s_t) = v_{\pi_z}(s_t) + (w_z - w_s)\overline{H}_{\pi_z}(a_t|s_t, z_t)$. Without loss of generality, $v_{\pi_z}(s_t)$ can be rewritten as $v_{\pi_z}(s_t) = v_{\pi_z}(s_t, z_t)$ because $z_t = f(s_t)$ is a deterministic function of $s_z$. Through the same procedure in Section 4.1.1 with a replacement $s'_t = (s_t, z_t)$, $J_{\pi_z}(s_t)$ is derived as

$$J_{\pi_z}(s_t) = \widetilde{v}_{\pi_z}(s'_t) = \mathbb{E}_{\pi_z}[\widetilde{q}_{\pi_z}(s'_t, a_t) + (w_z - w_s)h_{\pi_z}(a_t|s'_t)], \tag{12}$$

where $\widetilde{v}_{\pi_s}(s'_t)$ and $\widetilde{q}_{\pi_z}(s'_t, a_t)$ are the soft value function and soft Q-function for $\pi_z$ under the given $s'_t = (s_t, z_t)$, respectively.

Unlike the entropy term $w_s h_{\pi_s}(a_t|s_t)$ in Equation (10) which incentivizes the policy to have rich exploration, the entropy term $(w_z - w_s)h_{\pi_z}(a_t|s'_t)$ in Equation (12) does not guarantee that the policy has rich exploration. The variance of the policy's actions depends on whether $(w_z - w_s)$ is positive or not. For example, when $(w_z - w_s)$ is negative, the policy gets a penalty for actions with a high variance. Thus, the policy learned from Equation (12) has bounded exploration. The bounded exploration reduces an action variance such that the policy focuses on exploring around a given state and does not explore unnecessarily widely. In a perspective of experience data for learning, detailed experiences from $\pi_z$ can supplement widely obtained experiences from $\pi_s$.

### 4.2 Automating entropy adjustment

Because an entropy of policy is affected by not only its own distribution but also the type of task and the magnitude of rewards, the entropy of the policy can vary during training. For this reason, the temperatures $w_s$ and $w_z$ need to be set according to these considerations. As shown in [25], a temperature can be adjusted by optimizing dual problem for a constrained optimization problem.

First, the objective function in Equation (4) can be represented by soft Q-function as follows:

$$J_\pi(s_t) = \mathbb{E}_\pi[\widetilde{q}_\pi(s_t, a_t) + w_s h_{\rho_s}(a_t|s_t) + (w_z - w_s)h_{\rho_z}(a_t|s_t, z_t)], \tag{13}$$

where $\widetilde{q}_\pi(s_t, a_t) = r(s_t, a_t) + \gamma\mathbb{E}_{s_{t+1}\sim p}[\widetilde{v}_\pi(s_{t+1})]$ and $\widetilde{v}_\pi(s_t) = \mathbb{E}_\pi[\widetilde{G}_t]$ are the soft Q-function and soft value function for $\pi$, respectively.

For the objective function in Equation (13), a constrained optimization problem can be derived as

$$\max_\pi \; \mathbb{E}_\pi[\widetilde{q}_\pi(s_t, a_t)] \tag{14}$$

$$\text{s.t.} \; \mathbb{E}_\pi[h_{\rho_s}(a_t|s_t) - h_{\rho_z}(a_t|s_t, z_t)] \geq 0, \tag{15}$$

$$\mathbb{E}_\pi[h_{\rho_z}(a_t|s_t, z_t)] \geq 0. \tag{16}$$

For the constrained optimization problem, each entropy can be the bound condition to each other, i.e., $h_{\rho_s}(a_t|s_t)$ is the upper bound from the perspective of $h_{\rho_z}(a_t|s_t, z_t)$ and $h_{\rho_z}(a_t|s_t, z_t)$ is the lower bound from the perspective of $h_{\rho_s}(a_t|s_t)$. Because $h_{\rho_s}(a_t|s_t)$ does not have an upper bound, the policy is able to pursue diverse actions only using the state. Using the state and heuristic information, however, the policy is limited to get diverse actions due to the upper bound $h_{\rho_s}(a_t|s_t)$. Thus, for the case of the given state and heuristic information, the policy has to find balanced actions between the upper bound $h_{\rho_s}(a_t|s_t)$ and the lower bound 0.

A dual problem for the constrained optimization problem is defined as

$$\min_{w_s, w_z} \; \mathbb{E}_\pi[\widetilde{q}_\pi(s_t, a_t) + w_s h_{\rho_s}(a_t|s_t) + (w_z - w_s)h_{\rho_z}(a_t|s_t, z_t)], \tag{17}$$

where $w_s$ and $w_z$ are the Lagrange multipliers. The primal problem for $\pi$ and the dual problem for $(w_s, w_z)$ can be solved by using gradient descent method recursively. In practice, the soft Q-function is modeled by function approximation based on deep RL. Under the consideration of a parameterized soft Q-function for a fixed policy, $\widetilde{q}_\pi(s_t, a_t)$ does not affect to the Lagrange multipliers. Then, the optimal Lagrange multipliers for the given policy can be obtained by

$$w_s^*, w_z^* = \arg\min_{w_s, w_z} \; J_{\mathbf{w}}, \tag{18}$$

where $\mathbf{w} = [w_s, w_z]$ and

$$J_{\mathbf{w}} = \mathbb{E}_\pi[w_s h_{\rho_s}(a_t|s_t) + (w_z - w_s)h_{\rho_z}(a_t|s_t, z_t)]. \tag{19}$$

After obtaining the optimal Lagrange multipliers, the optimization for the policy is performed for the given Lagrange multipliers. Therefore, $\pi$ and $\mathbf{w}$ are updated by such a recursive optimization.

**Algorithm 1** EBE-AC
___
1: Initialize parameters $\theta_m$ and $\phi_m$ for $m \in \{s, z\}$.
2: Initialize temperatures $\mathbf{w}$.
3: Set: $\bar{\theta}_m \leftarrow \theta_m$ and $\mathcal{D} \leftarrow \emptyset$.
4: **for** each iteration **do**
5:     **for** each environment step **do**
6:         Select behavior policy index: $m_t \sim p_m(x)$
7:         Sample action: $a_t \sim \pi_{m_t}(a_t|s_t)$
8:         Step environment: $(s_{t+1}, r_t) \sim p(s_{t+1}, r_t|s_t, a_t)$
9:         Store a transition: $\mathcal{D} \leftarrow \mathcal{D} \cup (m_t, s_t, a_t, r_t, s_{t+1})$
10:     **end for**
11:     **for** each gradient step **do**
12:         Update soft Q-functions:
13:             $\theta_m \leftarrow \theta_m - \lambda \nabla_{\theta_m} J_{\theta_m}$ for $m \in \{s, z\}$ using Equation (20)
14:         Update policies:
15:             $\phi_m \leftarrow \phi_m - \lambda \nabla_{\phi_m} J_{\phi_m}$ for $m \in \{s, z\}$ using Equations (10) and (12)
16:         Update temperatures:
17:             $\mathbf{w} \leftarrow \mathbf{w} - \lambda \nabla_{\mathbf{w}} J_{\mathbf{w}}$ using Equation (19)
18:         Update target soft Q-functions:
19:             $\bar{\theta}_m \leftarrow \tau \theta_m + (1 - \tau)\bar{\theta}_m$ for $m \in \{s, z\}$
20:     **end for**
21: **end for**
___

### 4.3 Exploration-bounded exploration actor-critic algorithm

To solve the optimization problems in Section 4.1, actor-critic method with function approximation is adopted. For function approximations, parameterization method is applied to soft Q-function $\widetilde{q}_{\theta_m}(s_t, a_t)$ and policy $\pi_{\phi_m}$ for $m \in \{s, z\}$ in Equations (10) and (12). As shown in [25], the parameterized soft Q-function $\widetilde{q}_{\theta_m}(s_t, a_t)$ is trained to minimize the soft Bellman residual as

$$J_{\theta_m} = \mathbb{E}_{(s_t, a_t) \sim \mathcal{D}}\left[\frac{1}{2}\left\{\widetilde{q}_{\theta_m}(s_t, a_t) - \left(r(s_t, a_t) + \gamma \mathbb{E}_{s_{t+1} \sim p}[\widetilde{v}_{\bar{\theta}_m}(s_{t+1})]\right)\right\}^2\right], \quad (20)$$

where $\mathcal{D}$ is a replay buffer, $\widetilde{q}_{\theta_m}(s_t, a_t)$ is $\widetilde{q}_{\theta_s}(s_t, a_t)$ for $m = s$ and $\widetilde{q}_{\theta_z}(s'_t, a_t)$ for $m = z$,

$$\widetilde{v}_{\bar{\theta}_m}(s_{t+1}) = \begin{cases} \widetilde{q}_{\bar{\theta}_s}(s_{t+1}, a_{t+1}) + w_s h_{\pi_{\phi_s}}(a_t|s_{t+1}), & \text{for } m = s \\ \widetilde{q}_{\bar{\theta}_z}(s'_{t+1}, a_{t+1}) + (w_z - w_s) h_{\pi_{\phi_z}}(a_t|s'_{t+1}), & \text{for } m = z \end{cases},$$

and $\bar{\theta}_m$ is the parameters of a target soft Q-function obtained as an exponentially moving average of the soft Q-function parameters. After obtaining the parameterized soft Q-functions, the parameterized policies and temperatures are updated sequentially.

These updates are summarized in Algorithm 1. The soft Q-functions and policies are updated in the same manner with SAC [25]. As SAC uses two soft Q-functions to mitigate positive bias in a policy improvement, we also use two soft Q-functions for each policy. For the simplicity of Algorithm 1, we omit a detailed description of the two soft Q-functions for each policy.

## 5 Experiments

In order to validate our method, we implement a robotic manipulation task, *Pick-and-Lift*, modified from that of RLBench [26], where the task environment consists of a single UR3 arm with 6 DoF including the two-finger gripper, a cube object on a table, and a target position. We design the reward of *Pick-and-Lift* as the sum of dense and sparse rewards, in detail the rewards for reaching and moving sub-tasks are given by dense terms of distances among the robot's gripper, the cube and the target position, and the rewards for grasping is given by sparse terms of the binary signal according to whether or not to succeed in grasping.[5]

___
[5]Details of *Pick-and-Lift* are described in Appendix D.

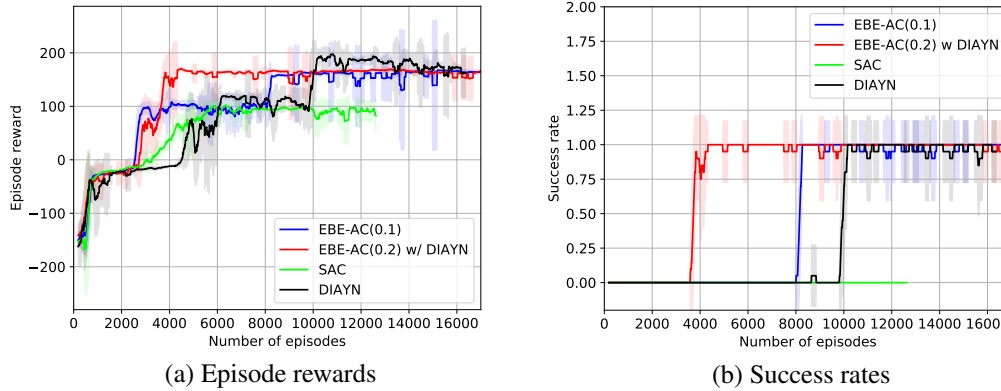

(a) Episode rewards      (b) Success rates

Figure 2: Training curves for *Pick-and-Lift* according to the number of episodes. In our experiments, the performance is measured by learning from scratch without any pre-trained models.

To demonstrate our method, we evaluate EBE-AC compared to SAC [25] and DIAYN [21], state-of-the-art methods in the field of stochastic policy based on deep RL. In addition, we evaluate EBE-AC with DIAYN to show effects of our method combining with latent information used in DIAYN.

## 5.1 Policy evaluation

We evaluate EBE-AC($\epsilon$) and EBE-AC($\epsilon$) combined with DIAYN, where $\epsilon$ is the probability defined in Equation (5).[6] Figure 2 represents training curves for episode rewards and success rates. In Figure 2a, a range of the episode reward in y-axis is roughly divided into three cases as follows:

- *Range 1*. (episode reward $< 0$): The robot tries to reach the cube.
- *Range 2*. ($0 <$ episode reward $< 100$): The robot tries to grasp the cube.
- *Range 3*. ($100 <$ episode reward): The robot tries to move the cube to the target position.

Compared with SAC and DIAYN, EBE-AC methods reach *Range 2* faster than the other methods. SAC can not reach *Range 3* and it means that SAC are stuck in local optimums *Range 2* during training. EBE-AC(0.1) and EBE-AC(0.2) with DIAYN methods reach *Range 3* faster than DIAYN. Especially, EBE-AC(0.2) with DIAYN is noticeably faster than DIAYN. [7]

In Figure 2b which represents the success rate, it is also observed that only the methods reaching *Range 3* have meaningful success rates. EBE-AC(0.2) with DIAYN outperforms the other methods and EBE-AC(0.1) has better success rates than DIAYN. Since EBE-AC is composed of the policy with exploration entropy and the policy with bounded exploration entropy, it can be seen as the combination of SAC's policy and the proposed policy with bounded exploration entropy, where the policies are linked by the temperature optimization. Through the composition characteristics of EBE-AC and experimental results for the fast convergences, therefore, it is shown that our method is able to help the prior works in terms of the sample efficiency.

## 5.2 Temperature Comparison

Figure 3 represents an analysis for temperatures according to the number of episodes,[8] where the temperatures of SAC and DIAYN are optimized in the same manner with SAC [25].

As shown in Figure 3, temperatures of all the methods decrease rapidly at the beginning of learning. The degradation of temperatures affects to reducing the proportion of entropy in a soft value function

---

[6]Empirically, we could gain the best performance for EBE-AC methods with the values $\epsilon = 0.1$ and $0.2$ for EBE-AC and EBE-AC with DIAYN, respectively. Results for various $\epsilon$ are described in Appendix D.5.

[7]Details of the episode rewards are described in Appendix D.6.

[8]For readability of Figure 3, we omit EBE-AC(0.2) with DIAYN because EBE-AC(0.1) is enough to explain the temperatures.

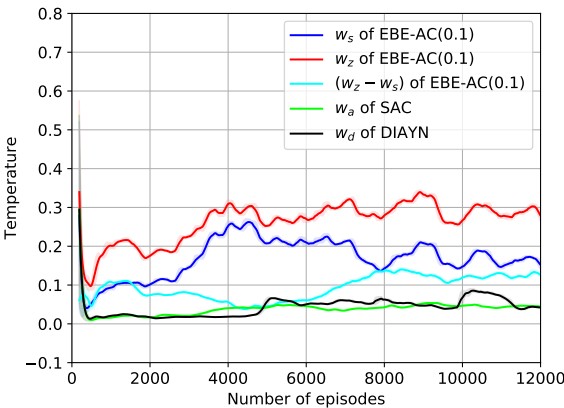

Figure 3: Compared to SAC and DIAYN, EBE-AC(0.1) utilizes the temperatures more dynamic than SAC and DIAYN.

which is a sum of soft Q-function and entropy. It means that the learning takes into account expected returns approximated by the soft Q-function more importantly.

The value of $w_z$ is always higher than that of $w_s$ and it leads to the value $(w_z - w_s)$ to be positive. Thus, the entropy term in Equation (11) affects $\pi_z$ to maintain exploratory. In addition, it is observed that $w_s$ in Equation (10) is higher than $(w_z - w_s)$. It means that $\pi_s$ is affected from the entropy and pursues more exploration than $\pi_z$. The temperatures of EBE-AC(0.1) increase rapidly at about 3000 and 8000 in x-axis, which is consistent with the increase of episode reward in Figure 2a. This feature is also observed for DIAYN at about 5000 and 10000 in x-axis. Therefore, EBE-AC(0.1) considers weighted entropies more than SAC and DIAYN, and is more exploratory than SAC and DIAYN. For this reason, EBE-AC(0.1) has better sample efficiency than SAC and DIAYN.

## 6 Conclusion

In this paper, we proposed exploration-bounded exploration actor-critic (EBE-AC), a novel deep RL algorithm that can optimize a stochastic policy through the use of heuristic information. The heuristic information includes interpretable human knowledge related to physical relationships between a robot and objects, which can be reused in various learning tasks. In a robotic manipulation task, we demonstrated that the proposed algorithm outperformed state-of-the-art off-policy actor-critic deep RL algorithms, such as SAC and DIAYN, in terms of sample efficiency. In addition, we extended the algorithm by means of latent information as DIAYN such that the extended algorithm can further improve sample efficiency and robustness.

**Acknowledgments**

This work was supported by Electronics and Telecommunications Research Institute (ETRI) grant funded by the Korean government. [21ZR1100, A Study of Hyper-Connected Thinking Internet Technology by autonomous connecting, controlling, and evolving ways]

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
