# OpenReview forum: "Stochastic Policy Optimization with Heuristic Information for Robot Learning"
_robot-learning.org/CoRL/2021/Conference — CoRL2021 Poster_

### Official Review · Reviewer_48sx · 2021-07-23

**Originality:** Good
**Technical Quality:** Good
**Clarity Of Presentation:** Good
**Impact:** 3

**Recommendation:**

Weak Accept: I recommend accepting the paper, but will not argue for my recommendation if the majority of other reviewers have a different opinion.

**Summary:**

The paper presents an information theoretic objective function to achieve both maximum entropy policy optimization and the maximum expected mutual information between the policy conditioned on heuristic information and unconditioned policy. The objective is then reformulated to two soft Q functions and all parameters are optimized in an actor critic way including two temperature parameters which control the degree of the maximum entropy and mutual information. In experimental results, the proposed method outperforms the state of the art soft actor critic algorithms like SAC and DIAYN in a pick-and-lift robot simulator.


**Issues:**

1. The experiments are not based on real robot systems, it might be better to transfer the existing result to a real robot system.
2. Figure 3-4 needs some improvement. It’s hard to tell the settings of each curve. And the legend should be consistent with all plots.

**Reviewer Expertise:**

Good: General knowledge of the area

**Strengths And Weaknesses:**

Pros:
    1. The information-theoretic objective takes human heuristic information into account which can guide exploration into unknown or a more reasonable region.
    2. There are two hyperparameters called temperature in the objective function. One encourages the actions to have rich exploration, the other one can encourage or bound exploration automatically. A dual problem is formulated to optimize the two temperature parameters.
    3. The experimental result is based on a pick-and-lift robot simulator. The performance of the proposed method outperforms existing soft actor critic method in both performance and speed of convergence.
Cons:
    1. The most interesting part of this paper is the temperature parameter which is optimized automatically. It would be great if the author can show the evolution of the temperature parameters during the training phase and explain why such results happen.
    2. Because this approach used step-based exploration rather than trajectory exploration. So it’s questionable whether there is temporally consistency in the robot motions, which is a big issue to apply this method on real robot tasks.

**Summary Of Recommendation:**

The paper presents an information theoretic objective function to achieve both maximum entropy policy optimization and the maximum expected mutual information between the policy conditioned on heuristic information and unconditioned policy. And two temperature parameters as well as the policy parameters are optimized automatically to encourage or bound the agent’s exploration automatically. Overall, this paper is marginally above the acceptance threshold.


### Post Rebuttal Comment:
I thank the reviewers for their comments. I keep my evaluation as is because the paper is missing real world experiments.

---

> ### Author Response · Authors · 2021-08-30
> **Response to Reviewer 48sx**
>
> -- Response to issues in Strengths And Weaknesses
>
> [R4-1: evolution of the temperature parameters] Thank you for your constructive comments. According to the reviewer’s comments, we added additional results for the evolution of the temperature parameters during the training phase in Section 5.2. As observed in Figure 3a, our method utilizes the temperatures more dynamic than SAC and DIAYN.
>
> [R4-2: applying real robots] We agree that the reviewer’s comments for that applying our method to real robot is a big issue. Final goal in our research plan includes dealing with real robot systems. In general, training real robots has a lot of cost, such as data acquisition time, unexpected risks, and etc. For this reason, we consider an approach as “Training in virtual environments and Applying in real worlds”. In more detail, we train policy models in virtual environments and apply these models to real robots. In other words, for real worlds, we do not use our learning methods but use our learned models, due to the learning costs.
>
> Currently, we are doing research on applying our learned model to a real robot. To this end, we have constructed a real robot testbed based on a virtual simulation, where real robot and virtual robot environments are synchronized by the ROS framework. To actuate the real robot in the testbed, we get joint positions of the virtual robot after one step based on the learned model, and send the joint positions to the real robot via ROS interface. Then, the real robot actuates own joints to the received joint positions via forward kinematics. Through these procedures, we have checked the feasibility of our learned model with real robot. Although we checked that the real robot acts similar to the virtual robot under the learned model, however, there is still a reality gap between the real and virtual robots, which sometimes leads to task failure of the real robot.
>
> To apply our results to the real robot robustly, further research is needed with respect to various considerations, such as adopting domain randomization, improving the testbed, utilizing real robot data, and etc. We think that it requires lots of efforts to solve the reality gap in robot manipulations, and hope that a novel solution would be presented in our future work.
>
>
> -- Response to issues in Summary Of Recommendation
>
> . Thank you for your positive comments. To further improve the paper, we did our best to reflect review comments in the revised version.
>
>
> -- Response to Issues
>
> [R4-3: applying real robots] We think that the Issue4-3 is addressed in details in “Strengths And Weaknesses”. Thus, please find the responses to Issue4-2.
>
> [R4-4: plot readability] To improve the readability of plots, we reformulated and modified plots including the legend. We left representative plots in the paper and moved the remaining plots to the supplementary materials.

---

### Official Review · Reviewer_oVi6 · 2021-07-23

**Originality:** Very Good
**Technical Quality:** Very Good
**Clarity Of Presentation:** Very Good
**Impact:** 4

**Recommendation:**

Strong Accept: I recommend accepting the paper and will argue for my recommendation even if other reviewers hold a different opinion.

**Summary:**

In this paper, the authors propose the "exploration-bounded exploration actor-critic" (EBE-AC) algorithm, a deep RL algorithm that can optimize a stochastic policy through the use of heuristic information. Such heuristic information includes prior human knowledge about physical relationships (e.g., between a robot and objects). The authors implement and evaluate their approach on a robotic manipulation task, Pick-and-Lift.

**Issues:**

* Can you point out some different domain/application examples?
* Improved plots would be good

**Reviewer Expertise:**

Very good: Comprehensive knowledge of the area

**Strengths And Weaknesses:**

PROS:
* The paper is well written and clearly structured
* The math is easy to follow
* The idea is interesting
* The experimental evaluation is sound and it includes the relevant baselines
* All the hyper-parameters and details are provided in the supplementary material for reproducing the experiments

CONS:
* This is not really a negative point, but it would be interesting to see another application of the proposed algorithm, eventually in a different domain.
* Plots could be improved for readability.

**Summary Of Recommendation:**

The paper is well written and well structured. The idea is clearly described by means of mathematical formulations, and the experimental evaluation is convincing. All the details for reproducing the experiments are provided in the supplementary material.

---

> ### Author Response · Authors · 2021-08-30
> **Response to Reviewer oVi6**
>
> -- Response to issues in Strengths And Weaknesses
>
> [R3-1: application in different domain]
> Thank you for your constructive and useful comments. As the reviewer’s comments, it is an important issue to apply our method to another domain and application such as drone navigation and autonomous driving. Although we presented our method in the domain of robot manipulation, we think that our method is well incorporated with theses other domains. Because the robot manipulation domain is similar to drone navigation and autonomous driving domains which deal with continuous action-state spaces. In addition, since various physical relationships can be designed in drone navigation and autonomous driving domains, these physical relationships can be easily utilized as heuristic information used in our method.
>
> [R3-2: plot readability] According to the reviewer’s comments, in order to improve the readability of plots, we reformulated and modified plots. We left representative plots in the paper and moved the remaining plots to the supplementary materials.
>
> -- Response to issues in Summary Of Recommendation
>
> . We appreciate your positive feedback.
>
>
> -- Response to Issues
>
> [R3-3] We think that the issues in this section are addressed in details in “Strengths And Weaknesses”. Thus, please find the responses to “Strengths And Weaknesses”.

---

### Official Review · Reviewer_UVE3 · 2021-07-24

**Originality:** Fair
**Technical Quality:** Fair
**Clarity Of Presentation:** Poor
**Impact:** 2

**Recommendation:**

Weak Reject: I recommend rejecting the paper, but will not argue for my recommendation if the majority of other reviewers have a different opinion.

**Summary:**

This paper presents a method for guiding exploration during reinforcement learning using binary and continuous heuristics.

**Issues:**

-- Please discuss the proposed work in more detail in relation to the works cited. What are the contributions of the proposed work? How does it advance the state-of-the-art?
-- The notation in the algorithmic sections needs to be revised; the p(m=[s|z]) notation in equations 5 and 7 is particularly strange
-- From equation 5, it seems that this is using an epsilon-greedy approach to exploration (guided by the heuristic) versus exploitation (guided by the current best policy).  If this is true, please just say that.
-- Consider moving some of the formalization from section 4 to the appendix (especially background information not critical to your contribution) and providing more detail on the heuristics used in the experiments
-- The charts in Figure 3 are very difficult to read and interpret
-- It would be helpful to give an intuition for what you're doing in section 4, especially where and how the heuristics come in; right now it's very difficult to follow
-- The entire first paragraph can be removed

**Reviewer Expertise:**

Good: General knowledge of the area

**Strengths And Weaknesses:**

Strengths
-- Incorporating human knowledge into learning is an important area of research
-- The authors have included significant formal detail about their approach

Weaknesses

-- The related work section is highly inadequate.
-- The paper was very difficult to read and needs significant proofreading
-- The evaluation fails to provide strong justification for the proposed method

Suggestions for Improvement

-- Include other methods that incorporate human knowledge into learning as baselines
-- Validate the method across more than task (ideally including real-world robot experiments as well as simulation)
-- Provide significantly more context to relate the contribution to prior work and the state-of-the-art


**Summary Of Recommendation:**

The general idea of using heuristics to speed learning (especially in robotics) is good, but the paper needs significant revisions to make it easier to follow, better situate the work relative to prior work, and more fully demonstrate that the approach improves over prior work.

---

> ### Author Response · Authors · 2021-08-30
> **Response to Reviewer UVE3**
>
> -- Response to issues in Strengths And Weaknesses
>
> [R2-1: related work section] Thank you for your constructive comments. We think that our main contribution is the utilization of human knowledge for reinforcement learning, especially stochastic policy learning. The state-of-the-art methods, SAC and DIAYN, are utilizing rich exploration to obtain various experiences at the beginning of learning, and tend to reduce the exploration as the learning progresses in order to acquire good expected returns stably. However, when a policy is stuck in inevitable local optimums which must be passed to complete a given complex task, the policy needs to utilize rich exploration in order to escape the local optimums even in the middle of learning. In other words, it is necessary that a policy simultaneously pursues rich exploration and stable expected returns throughout the learning period. A method using two types of stochastic policies is proposed in order to maintain rich exploration throughout the learning period. For this reason, we focus on improvement of the sample efficiency for stochastic policy learning in related work section. To emphasize our purpose, we modified Section 2.
>
> [R2-2: proofreading] According to the reviewer’s comments, we have done additional proofreading to improve the paper.
>
> [R2-3: evaluation] In the submitted version, we tried to describe too many things within the 8pages limitation. We agree that it is hard to focus on justification of our proposition. According to the reviewer’s comments, we modified the evaluation section via simplifying plots, moving redundant descriptions to Appendix, and emphasizing main contributions.
>
> [R2-4: baselines] As the reviewer’s comments, our proposition can be seen as one of the researches in learning from human. Learning from human is classified in inverse reinforcement learning(IRL), where an agent learns from well-defined data given by human experts. Thus, IRL approaches depend heavily on the data from human. However, reinforcement learning(RL) approaches are basically based on trial-and-error, where an agent learns from a lot of own experience data. Thus, sample efficiency is the most important issue in RL approaches.
>
> Main framework of our proposition follows RL approaches so that we tried to improve the sample efficiency of stochastic policy learning by using human knowledge. In other words, the human knowledge in our proposition is utilized to improve not “learning from human” in IRL but “learning from scratch” in RL. Because RL and IRL have different learning principles, it is not easy to compare each other under the same performance metrics. For this reason, we adopted baselines as the state-of-the-art methods in stochastic policy works as SAC and DIAYN, not in IRL fields. According to the reviewer’s comments, although we tried to find existing verified stochastic policy methods incorporating with heuristics, it is hard to find suitable works to compare with our method. However, SAC and DIAYN are well-known methods as the state-of-the-art so that we made efforts to present advances of our method in comparisons with such two methods.
>
> [R2-5: verification with additional task] In order to validate our method, we added another example task “Pick-and-Drag” in Appendix E, in which there are a single UR3 arm, a cube object, a stick, and a target position. A success condition of the task is “the cube is placed at the target position”. To success the task, the robot arm should pick up the stick and drag the cube by using the stick, because the length of the arm is not sufficient to move the cube to the target position. Because the interaction between the stick and cube exists in the task, “Pick-and-Drag” is more difficult than “Pick-and-Lift”. For the additional task, we also designed a reward and heuristic information in the same manner of “Pick-and-Lift”. Through the additional experimental results, we confirmed that the proposed method outperforms the baselines in terms of the sample efficiency for not only “Pick-and-Lift” but also “Pick-and-Drag”.
>
> As the reviewer’s comments, it may be the best way to validate our method with a real robot. Currently, we are doing research on applying our learned model to a real robot. To this end, we have constructed a real robot testbed based on a virtual simulation, where real robot and virtual robot environments are synchronized by the ROS framework. We have checked the initial feasibility of our learned model with the real robot. However, there is still a reality gap between the real and virtual robots, which sometimes leads to task failure of the real robot. We think that it requires lots of efforts to solve the reality gap in robot manipulations, and hope that a novel solution would be presented in our future work.

---

> > ### Author Response · Authors · 2021-08-30
> > **Response to Reviewer UVE3**
> >
> > -- Response to issues in Summary Of Recommendation
> >
> > [R2-6: paper improvement] We think that the paper improvement issues in Summary Of Recommendation are addressed in details in “Strengths And Weaknesses” and “Issues” sections. Thus, please find the responses to “Strengths And Weaknesses” and “Issues” sections.
> >
> >
> > -- Response to Issues
> >
> > [R2-7: contributions] We think that our contributions are three advantages.
> > 1. We designed a novel information theoretic metric in order for EBE-AC to have high dependencies between actions and heuristic information.
> > 2. We proposed epsilon-exploration based policy using two types of stochastic policies. (A difference with epsilon-greedy is described in R2-9
> > 3. Final, we proposed a temperature optimization to balance between exploration and bounded exploration policies.
> >
> > To clarify our contributions, we emphasized advantages of our method in the last paragraph in Section 1.
> >
> > [R2-8: notations] To clarify Equations 5 and 7, we added descriptions for the epsilon-exploration method and redefined p(m=s or z) with a modified notation and an exact naming as probability mass function.
> >
> > [R2-9: difference with epsilon-greedy] In comparison with an epsilon-greedy method which uses policy-based actions and random actions with epsilon probability for exploration effects, the epsilon-exploration method uses two types of policies having different exploration characteristics measured in entropies, where one learns to maximize a return with rich explorations and the other learns to maximize a return with bounded explorations. To clarify it, we added the description in footnote 2.
> >
> > [R2-10: moving formalizations to appendix] According to the reviewer’s comments, in order to simplify the paper, some formalizations in Section 4 are moved to the Appendix A.
> >
> > [R2-11: details on the heuristics] Since heuristic information can be designed flexibly by human experts (e.g. task designers), we did not present specific guides for heuristic information design, but present concepts in Section 3.2 and detailed examples used in the experiments in Appendix D.3.
> >
> > [R2-12: plot readability] To improve the readability of plots, we reformulated and modified plots. We left representative plots in the paper and moved the remaining plots to the supplementary materials.
> >
> > [R2-13: intuition of proposition] Consider the state-of-the-art, our method has advantages for efficiently collecting experience data for learning. In a perspective of experience data for learning, detailed experiences from pi_z can supplement widely obtained experiences from pi_s such as SAC and DIAYN. It means that our method can focus more on around of a given state than the baselines to solve tasks. To clarify intuition of our proposition, we modified descriptions in Section 4.1.2.
> >
> > [R2-14: remove first paragraph] According to the reviewer’s comments, we removed the first paragraphs in Sections 3 and 4 in order to simplify the paper.

---

### Official Review · Reviewer_p2oA · 2021-07-24

**Originality:** Fair
**Technical Quality:** Fair
**Clarity Of Presentation:** Good
**Impact:** 2

**Recommendation:**

Weak Reject: I recommend rejecting the paper, but will not argue for my recommendation if the majority of other reviewers have a different opinion.

**Summary:**

The authors propose to augment the observation with heuristic information in deep reinforcement learning.
Because the heuristic information might not always help the agent, they are learning two policies at the same time.
One is using the heuristic information while the other is not.
They performed experiments with ablation studies showing that their approach was learning faster.


**Issues:**

Some effort for the readability of the curves should be made.

Typos:
-line 105 are induced
-line 151 need
-line 240 maintain
-line 252 improve

**Reviewer Expertise:**

Excellent: Expert knowledge on the topic of the paper

**Strengths And Weaknesses:**

Strength:
The paper is easy to understand and well written.

Weaknesses:
The approach is a bit ad hoc. There is no real justification for picking one or the other policy.

In Fig. 5 DIAYN w/ z and EBE-AC(0.2) w/ DIAYN are actually quite close.
I suspect that the small gain of EBE-AC(0.2) w/ DIAYN is actually due to the fact that some information in z impedes the learning.
For instance, if I have not grasped the object yet f_distance(object, target position) and f_fingers will not help to make a decision.
Accordingly, once the object is already grasped, f_distance(gripper, object) is not helping.

Two missing baselines would be to learn two policies with SAC both without z and both with z. To see if the gain is not coming from a higher expressive power (2 policies).


**Summary Of Recommendation:**

I am not convinced by the method because it somehow tries to tackle an ill-defined problem.
Using this method, it is assumed that the information in z is not always useful. In this case, it would be easier to detect which information of z is useful at which times and condition the policy on that. Rather than learning two policies at the same time, it seems easier to learn a succession of policies. For example, one to catch the object with a well-defined z, then another to move it to the target with another z definition.

---

> ### Author Response · Authors · 2021-08-30
> **Response to Reviewer p2oA**
>
> -- Response to issues in Strengths And Weaknesses
>
> [R1-1: small gain] Thank you for your constructive comments and suggestions. In this work, we focused on obtaining better sample efficiency, and we obtained the result that our proposed EBE-AC(0.2) w/ DIAYN can reduce about 1500 episodes for the convergence level than DIAYN w/ z, as shown in Fig. 5. We think that this gap is a meaningful result in terms of sample efficiency.
>
> [R1-2: effect of z] As shown in the experiment results for DIAYN w/ z and SAC w/ z, DIAYN w/ z and SAC w/ z have better convergences than the methods without z. It means that the heuristic information z affects the convergences.
>
> [R1-3: baselines] We added the baselines both SAC with z and without z in Appendix D.4 For readability of plots, we omitted the baseline SAC with z in Figure 2.
>
>
> -- Response to issues in Summary Of Recommendation
>
> [R1-4: usefulness of z] Even though the reviewer worries about the usefulness of the z, we are convinced that the information z is useful to maximize the value functions over the entire learning episodes. In the perspective of value function which represents an expected return at the end of an episode, the z as the input parameters of value function affects not only the current step but also steps over the entire episode. It means that the information z is useful for all policies utilizing z to make decisions that maximize the value functions over the entire episode. In the submitted version, the information z was not explicitly described as an input of value function for the policy pi_z. To avoid confusion and clarify the usage of the information z, we modified the representation of the value function for the policy pi_z in Section 4.1.2.
>
> [R1-5: learning a succession of policies] As the reviewer’s suggestions for learning a succession of policies, we also considered the approach identical to the suggestions in the initial study. The learning a succession of policies can be seen as learning primitive skills separately. If the number of primitive skills is two, e.g. “Reach-and-Push”, the learning primitive skills may be more effective than our method in terms of sample efficiency. However, if the number of required primitive skills is larger than two, the learning primitive skills would be not effective than our method. For example, task “Pick-and-Lift” requires at least three skills, such as reaching, grasping, and moving (it requires to keep grasping). In the learning primitive skills approach, the number of polices increases according to the number of primitive skills, where multiple replay buffers are required for gathering different experiences for the primitive skills. In addition, to utilize the learned primitive skills effectively, a high-level policy or a meta-policy is needed, which requires additional learning processes. Thus, the learning primitive skills approach requires a complicated implementation with multiple pre-training phases for primitive skills and a meta-training phase for the high-level policy, which leads to degrade sample efficiency and to limit learning from scratch.
>
> In contrast, our method requires only two policies and shares a single replay buffer. Owing to its simple structure, it is easier to implement than the learning primitive skills approach. In addition, because our method is able to learn from scratch, it does not require pre-training and meta-training phases, which causes a decrease in sample efficiency.
>
>
> -- Response to Issues
>
> [R1-6: plot readability] According to the reviewer’s comments, in order to improve the readability of plots, we reformulated and modified plots. We left representative plots in the paper and moved the remaining plots to the supplementary materials.
>
> [R1-7: typos] According to the reviewer’s detailed comments, we modified all the typos existing in the paper.

---

> > ### Comment · Reviewer_p2oA · 2021-09-03
> > **Author Response Acknowledged**
> >
> > I thank the authors for their answer.
> >
> > The missing baselines I was mentioning would still learn *two* policies and use your stochastic epsilon-greedy selector.
> > It is a way to verify if your performance increase does not come from the fact that you are learning two policies at the same time.
> >
> > Regarding the usefulness of z, you did not really address my concerns. If z is always useful, then it should always be provided.
> > Given that you are learning a policy without z, it means that it is not always useful.
> > Hence, I think the major question is when to add z or not and which components of z should be added when. However, in your approach, you're only using an epsilon-greedy selection.
> >
> > I think your method somehow suffer the same issue as a succession of policies. If you split z into two vectors z1, z2 and learn 3 policies: one with z, one with z1, one without any z, it could maybe improve the convergence speed even more. Can we really be sure that taking z as its whole is the best way to improve the learning speed?

---

> > > ### Author Response · Authors · 2021-09-10
> > > **Response to Reviewer p2oA**
> > >
> > > [C1-1: missing based lines] We agree the reviewer’s concerns for the performance verification. We think that the performance increase of our method comes from not only learning two policies but also optimizing temperatures which balances the entropies for the two policies. We added additional results for the evolution of the temperature parameters during the training phase in Section 5.2. As observed in Figure 3, our method utilizes the temperatures more dynamic than SAC and DIAYN. Therefore, our method considers weighted entropies more than SAC and DIAYN, and is more exploratory than SAC and DIAYN. We think that the epsilon selector method for SAC and SAC with z, which is a simple selector without optimizing temperatures, obtains marginal gains compared with our method.
> > >
> > >
> > > [C1-2: usefulness of z] As the reviewer’s comments, the usefulness of z is dedicated to the policy with bounded exploration, pi_z(a|s,z) in Section 4.1.2. Although z in our method is designed by human experts (e.g. task designers), z is not explicitly utilized as symbolic AI does, but implicitly utilized by deep learning approaches. Thus, we did not determine when to add z and which components of z to use, but utilize z as the input of deep neural networks such as actor(policy) and critic(value function) networks. The actor-critic networks extract automatically useful information of each element of z combined with the states in a deep learning manner. In other words, deep neural networks utilize z combined with the states as input and extract its useful information automatically. In the perspective of critic network, the value function implemented by deep neural network is the approximator for an expected return, which are trained by Temporal Difference(0) method. For simple example of Pick-and-Lift, let z=1 and z=0 indicate that the robot grasped the cube and does not, respectively. In the perspective of human as symbolic AI, it is easy to infer that the robot with z=0 should try to grasp the cube and the robot with z=1 should move the cube to the target position. However, in the perspective of critic network of DRL, the value function cannot critic which is good between z=0 and z=1 at the beginning of value function training. From a well-trained value function, it is only given that robot with z=0 has low expected return and robot with z=1 has high expected return. It means that z is always useful for the value function to critic whether a state is good or not over all states in episodes. If a given state has low expected return, the actor network tries to reach a state which has high expected return.
> > >
> > > In addition, z affects to our method widely and implicitly. Our method based on off-policy learning with a replay buffer is summarized simply as collecting trajectories in the replay buffer, learning two critic networks, learning two actor networks, and optimizing temperatures. Through the epsilon selection method, trajectories acquired by pi(a|s,z) are stored in the replay buffer. The trajectories are used to learn not only actor-critic networks of pi(a|s,z) but also actor-critic networks of pi(a|s). In addition, the trajectories are used to optimize the temperatures. It means that z affects to not only the trajectories in the replay buffer but also learning and optimizing procedures implicitly. As we mentioned above, because deep neural networks extract useful information of z automatically combined with the states, it is hard to measure effects of z explicitly in our method. Through the experimental results, we just observed that the policy with z has different entropy compared with the policy without z, then different trajectories are acquired by the policy with z. Finally, various trajectories in the replay buffer, which are acquired by both the policies with and without z, lead to improvements in the sample efficiency of trainings. This observation is related to [R1-2: effect of z]. Owing to DRL based-approach, it is hard to present the usefulness of z explicitly, but z is useful implicitly in terms of the sample efficiency.

---

> > > > ### Author Response · Authors · 2021-09-10
> > > > **Response to Reviewer p2oA**
> > > >
> > > > [C1-3: learning a succession of policies] As we answered in [C1-2], neural networks extract useful information of z automatically combined with the states. So, if we design three networks for different primitive skills, we will use all elements of z for each network. We think that useful information of z is automatically extracted combined with the states and utilized to each network for a given primitive skill in a deep learning manner. Compared with our method, using individual networks for each primitive skill could be better than our method according to how optimized it is. In the answer [R1-5], we compared primitive skill-based learning with our method in terms of implementation structure such as the number of networks, the number of replay buffers, pre-training phase, and meta-training phase. To compare performance with primitive skill-based learning with z, optimizing the primitive skill-based learning with z should be studied, which will be a new research issue. For this reason, we verified performance of our method compared with well-known baselines as SAC and DIAYN.

---

### Meta-Review · Area_Chair_Jpx4 · 2021-08-13

**Recommendation:** Accept (Poster)
**Confidence:** 4

**Metareview:**

The paper proposes a new approach for policy optimization. The main contribution lying at the information theoretic objective function that targets to achieve both maximum entropy policy optimization and the maximum expected mutual information between the policy conditioned on heuristic information and unconditioned policy. It is suggested that there should be more details on the choice of  the heuristics used in the experiments. And it would be great if there are additional validation results  across more tasks or discussions on the use of the proposed formulation for different domain/application examples.

After the rebuttal: The authors have made great effort in providing additional results and clarifying concerns from the reviewers. There is one critic that the proposed method has addressed an ill-defined problem, which I think can be an interesting direction for the authors to look at for future researches. On the other side, the proposed method still makes sense and is accompanied with a good set of experimental results that demonstrate some of the claims.

---

> ### Author Response · Authors · 2021-08-30
> **Response to Metareview**
>
> We wish to thank the area chair and reviewers for their careful review and comments on our paper. To improve the quality of our paper, we have made several revisions according to the reviewer’s comments.
> The major modifications in the revised version are as follows:
> - To emphasize our contribution points in Sections 1, 2, 3 and 4.
> - To add an analysis for temperatures in Figure 3.
> - To give details on the heuristic information in Appendix D
> - To add an additional example task “Pick-and-Drag” in Appendix E.
> - To enhance the readability of plots.
>
> We did our best to answer all issues addressed from reviewers. Please find the reply to the reviewer’s comments below. Thank you for your advices again.

---

### Decision · Program_Chairs · 2021-09-13

**Decision:**

Accept (Poster)

**Comment:**

The paper proposes a new approach for policy optimization. The main contribution lying at the information theoretic objective function that targets to achieve both maximum entropy policy optimization and the maximum expected mutual information between the policy conditioned on heuristic information and unconditioned policy. It is suggested that there should be more details on the choice of  the heuristics used in the experiments. And it would be great if there are additional validation results  across more tasks or discussions on the use of the proposed formulation for different domain/application examples.

After the rebuttal: The authors have made great effort in providing additional results and clarifying concerns from the reviewers. There is one critic that the proposed method has addressed an ill-defined problem, which I think can be an interesting direction for the authors to look at for future researches. On the other side, the proposed method still makes sense and is accompanied with a good set of experimental results that demonstrate some of the claims.